# Acquired Haemophilia A: A 15-Year Single-Centre Experience of Demography, Clinical Features and Outcome

**DOI:** 10.3390/jcm11102721

**Published:** 2022-05-11

**Authors:** Raisa Guerrero Camacho, María Teresa Álvarez Román, Nora Butta Coll, Damaris Zagrean, Isabel Rivas Pollmar, Mónica Martín Salces, Mercedes Gasior Kabat, Víctor Jiménez-Yuste

**Affiliations:** 1Servicio de Hematología y Hemoterapia, Hospital Universitario La Paz, Paseo de la Castellana 231, 28046 Madrid, Spain; raisa.guerrerocamacho@nhs.net (R.G.C.); damariszagrean01@gmail.com (D.Z.); mirivas718@gmail.com (I.R.P.); monicamsalces@gmail.com (M.M.S.); mer501@hotmail.com (M.G.K.); vjyuste@gmail.com (V.J.-Y.); 2Coagulopathies and Haemostasis Disorders Group, IdiPAZ, Paseo de la Castellana 231, 28046 Madrid, Spain; 3Facultad de Medicina, Universidad Autónoma de Madrid, Arzobispo Morcillo, 4, 28029 Madrid, Spain

**Keywords:** acquired, haemophilia A, demography, presentation

## Abstract

Acquired haemophilia A (AHA) is a rare severe bleeding disorder resulting from the production of autoantibodies directed against coagulation factor VIII. At presentation, bleeding events can be severe, and an early diagnosis and treatment are of major importance. The current study aims to analyse the treated patients who have been diagnosed with AHA for a better understanding of our population and treatment outcome. We conducted a retrospective study with 26 patients who had been diagnosed with AHA and who were treated in our hospital between January 2006 and January 2021. The patients ranged in age from 30 to 85 years old: 46.10% were men, 46.10% had no known underlying condition, 27% had an underlying malignancy, 7.60% presented with other diseases: psoriatic arthritis and Paget’s disease, and 19.30% presented with AHA during puerperium. All of the patients had bleeding events and were treated with bypass agents for this as well as with immunosuppressive therapy to eradicate the inhibitor. A total of 53.80% of the patients had major bleeding. Sixty-nine percent of the patients achieved complete remission, but 26.90% died during the follow-up, although bleeding was not the cause of death in any of these cases. Our observations underline the importance of clinical suspicion and early referral to centres with experience and laboratory facilities for managing AHA.

## 1. Introduction

Acquired haemophilia A (AHA) is a rare disease that is caused by the production of autoantibodies against factor VIII (FVIII) in patients with no previous history of coagulopathy. Clinically, AHA presents with bleeding that can be either spontaneous, provoked, or both and is life-threatening [1,2,3,4,5,6]. The incidence varies between 0.10 and 1.48 per million inhabitants per year [2,5,6,7,8,9,10,11,12,13,14,15,16] and predominantly affects older people [16,17,18], with the likelihood of the condition increasing with age and being very rare in children [19,20,21]. Although AHA is idiopathic in most cases, an association with underlying conditions has been reported, most often in malignancies, during puerperium, and in autoimmune diseases [16,17]. Over the years, observational studies have provided a better understanding of the clinical features, demographics, and treatment response for AHA. The United Kingdom Haemophilia Centre Doctor’s Organisation performed a prospective surveillance study in 2001–2003 and presented the results in 2007 with a complete description of the features of AHA and the outcomes [16]. In 2009, Huth-Kühneet et al., published international recommendations on the diagnosis and treatment of patients with AHA based on comprehensive literature searches and the experience of the authors [18]. In 2012, the European Acquired Haemophilia Registry 2 (EACH2) presented a large scale, pan-European database that included 13 European countries and 117 separate treatment centres, collecting extensive data on demographics, diagnosis, underlying disorders, bleeding characteristics, and haemostatic and immunosuppressive therapies [17,19,20,21,22]. The prospective French registry Surveillance des Auto antiCorps au cours de l’Hémophilie Acquise (SACHA) collected data on the prevalence, clinical course, disease associations, and treatment outcomes [23]. The treatment decision is often based on the treating clinician’s expertise. A consensus was reached when an international guideline was published in 2020 that provided recommendations based on registry findings, experience, and the knowledge of the authors. To better understand the disease in our population, this study presents a retrospective analysis of a single-centre experience in managing AHA over a period of 15 years.

## 2. Materials and Methods

La Paz University Hospital received 26 patients with AHA between January 2006 and January 2021. Data were collected retrospectively from the patients’ medical records. The patients were diagnosed based on an acute or recent onset of bleeding, a prolonged activated partial thromboplastin time (aPTT), and reduced FVIII activity. Inhibitor detection screening was performed with the mixing test conducted using the aPTT method, which was considered positive if the aPTT of the mixed normal and patient’s plasma (1:1) incubated for 2 h persisted as prolonged [24]. The AHA diagnosis required plasma FVIII activity <50 U/dL and the detection of the FVIII inhibitor, whose titre was determined using the chromogenic FVIII assay and Bethesda method [25]. The local laboratory evaluated the aPTT, FVIII:C activity, and inhibitor titre. Inclusion criteria: diagnosis of AHA. Exclusion criteria: previous history of a bleeding disorder.

Patients were investigated for underlying conditions via computerized tomography of the chest, abdomen, and pelvis as well as via their autoimmune profile.

Major bleeding was defined as intrabdominal bleeding, bleeding in the central nervous system or a transfusion requirement of three or more units of red blood cells.

Complete remission was defined as an FVIII level of >50 U/dL, undetectable FVIII inhibitor [<0.6 Bethesda unit (BU)], and immunosuppression that was stopped or reduced to the doses used before acquired haemophilia developed without relapse.

### Statistical Analysis

The continuous quantitative variables were described using the median and interquartile range (IQR), and the qualitative variables were described using absolute and relative frequencies expressed as percentages. When deemed appropriate, we reported the minimum and maximum values. Comparisons between the continuous quantitative variables between separate groups were primarily rated by non-parametric tests, such as the Kruskal–Wallis or Mann–Whitney U tests. The frequency analysis between the qualitative variables was performed by testing the chi-squared value with the Yates’s correction or Fisher’s exact test. *p*-values of <0.05 were considered statistically significant. Time to remission and survival time were studied using the Kaplan–Meier method with the log-rank test for the qualitative variables. The data were processed by a computer using a database in Microsoft Excel format, which was later imported into SAS version 9.4 (SAS Institute Inc. 2013, SAS base^®^ 9.4 SAS/STAT—Statistical analysis; Cary, NC, USA) for statistical processing.

## 3. Results

### 3.1. Age and Sex

The median age was 66 years old (IQR 55–81), with patients ranging from 30 to 85 years of age. Just under half of the patients were men (46.10%) who were older than 70 years of age with a minimum age of 52 years at presentation, whereas 19.30% of the women were younger than 40 years (Figure 1).

### 3.2. Underlying Disease

Twelve (46.10%) patients had AHA but no known or proven underlying condition (idiopathic); nine (75%) of these patients were older than sixty-five years of age. AHA appeared during puerperium in five (19.23%) patients and was the most frequent underlying condition in the patients who were younger than 65 years of age. Seven (26%) patients had an underlying malignancy that was known to the patient before the diagnosis of AHA; a total of 85.70% of the patients with an associated malignancy were older than 65 years of age. Two (7.69%) patients presented with other diseases: psoriatic arthritis and Paget’s disease, and these diseases were distributed equally (50%) to those older and younger than sixty-five years of age (Figure 2).

Neoplasms were more frequent in men older than 65 years old (*p* = 0.01), puerperium was, of course, only present in women, and the other underlying conditions had a similar distribution among the male and female patients.

### 3.3. Bleeding Presentation

All patients had some type of bleeding; 96% had a haematoma during disease progression, and only one (4%) patient presented with no haematoma but did have gastrointestinal and urinary bleeding. Table 1 summarises the bleeding distribution. There was no association between bleeding distribution or spontaneous presentation and the underlying disease, age, or sex (*p* > 0.05).

### 3.4. Laboratory

Table 2 summarises the laboratory data. The minimum and maximum haemoglobin level at diagnosis was 6.80 g/dL and 14.40 g/dL, respectively, with aPTT of 41.90–123.50 s, with no statistically significant association according to sex (*p* > 0.05).

The inhibitor titre range at diagnosis was 0.70–972 BU. One of the patients was not analysed because they were the result of an interhospital transfer and did not have the first inhibitor titre reported in their medical records. The inhibitor titre for the female patients presented with a median of 7.75 BU (IQR 4.55–17.30 BU), while the men had a median of 21.90 BU (IQR 7.90–108.90 BU).

FVIII:C activity at diagnosis was found to be between 0 and 5 U/dL and was undetectable in 61.53% (16) of the patients and within the range of 0.20 U/dL to 5 U/dL in 38% (10) of the remaining patients.

No statistically significant association was found between the inhibitor titre with age or sex (*p* = 0.92, *p* = 0.05) and between FVIII at diagnosis and age or sex (*p* = 0.46, *p* = 0.91).

### 3.5. Transfusion Requirement

Sixteen (61.50%) patients required transfusion support with 2–56 units of red blood cells (median 6 units, IQR 3.50–14). The transfusion requirements were more frequent for the patients who were older than 65 years old (42.30%), although this association was not statistically significant. The same proportion of men and women required transfusion (eight patients in each group). Likewise, there was no statistically significant association (*p* = 0.37) between the transfusion requirements and the underlying disease.

### 3.6. Major Bleeding

Fourteen (53.80%) of the patients presented with major bleeding (Table 3). Of these, only one was not administered three or more units of concentrate and experienced intra-abdominal bleeding.

Major bleeding was not associated with sex, underlying disease, or age group (Table 3). There was no statistically significant difference between the mean levels of the inhibitor titre at diagnosis and factor VIII between the patients who experienced major bleeding and those who did not (Figure 3).

### 3.7. Treatment

Patients received haemostatic treatment with bypassing agents or human FVIII during bleeding episodes (Table 4), as well as immunosuppressive treatment to eradicate the inhibitor.

#### Immunosuppressive Therapy (IST)

All of the patients received prednisolone (1–1.5 mg/kg/d) or an equivalent steroid as part of their inhibitor eradication therapy; 84.60% of the patients received rituximab (375 mg/m^2^ weekly for 4 weeks), 73.10% received at least one dose of intravenous immunoglobulins (0.4 g/kg/d, 5 days), and seven (26.90%) patients received another immunosuppressant, such as cyclosporine, cyclophosphamide, or mycophenolate.

### 3.8. Response to Treatment

Eight patients did not achieve complete remission and were in the 65-and-over age range; however, the difference was not statistically significant (*p* = 0.136). Four of these patients continued on IST (cyclophosphamide, ciclosporine, or micophenolate) and had normal FVIII:C activity and an undetectable inhibitor titre, relapsing on discontinuation attempts; two of them died from complications resulting from an infection; two were alive at the end of the follow-up. Two patients died while on IST with FVIII:C < 50 dL. IST was discontinued in one patient with poor performance status, who died while on best supportive care management. One patient on IST was lost at follow-up.

Sixty-nine percent of the patients achieved complete remission. All of the patients under 65 years of age achieved complete remission. Complete remission was more frequent in the women than it was in the men (12 women and 6 men), with no statistically significant association being observed (*p* > 0.05).

The median time to complete remission was 55 days (IQR 45–65), with a minimum of 33 days and a maximum of 143 days.

No statistically significant association was observed between complete remission and age (*p* = 0.52), haemoglobin levels (*p* = 0.71), aPTT (*p* = 0.66), or FVIII (*p* = 0.56). The patients who did not achieve complete remission had high inhibitor levels at presentation (*p* < 0.05).

### 3.9. Complications

Two (7.70%) patients presented with thrombotic complications, four (15.40%) presented with infectious complications, and two (7.70%) presented with other types of complications, such as surgical complications (perforation), pleural effusion, and iatrogenic pneumothorax. All of these complications were observed in the patients who were older than 65 years old.

Complications were more frequent in men who presented all the thrombotic events (two patients), 75% of infectious patients (three patients), and all other complications (two patients).

### 3.10. Survival

After AHA diagnosis, the mean survival time was determined to be 139.20 (95% CI 105.80–174) days (Figure 4). Seven (26.90%) patients died during the study, and two of them died in complete remission for their underlying malignancy (bladder cancer and melanoma, respectively). One patient died from bowel cancer without achieving complete remission. Two patients died due to infection with normal levels of FVIII and an undetectable inhibitor titre, one of whom was being IST dependent. One of these patients died from severe acute respiratory syndrome coronavirus 2 (SARS-CoV-2) infection and pseudomonas aeruginosa, and the other died from sepsis of an unknown origin. One patient died from multi-pathogen sepsis caused by listeria, clostridium, and staphylococci while haemostatic and after IST had been discontinued. One patient did not achieve complete remission and died on best supportive care, including haemostatic treatment.

Five of the seven deceased patients were older than 65 years of age and male. There was no statistically significant association between death and age (*p* = 0.69) or sex (*p* = 0.03). Six (85.70%) of the deceased patients required transfusion support; no association was found between death and transfusion requirements (*p* = 0.19). A comparison of survival in patients according to age and transfusion requirement is shown by the Kaplan-Meier plot in Figure 5. Five (71.40%) of the deaths occurred in patients with underlying malignant diseases; the rest of the patients had no underlying condition, and the results were statistically significant (*p* = 0.02). Comparison of survival according to underlying disease is shown in the Kaplan-Meier plot Figure 6. No death was reported in the patients who only presented with provoked bleeding. Of the deceased patients, only six had spontaneous bleeding, and one had spontaneous and provoked bleeding. These results were not statistically significant (*p* = 0.40). Urinary bleeding occurred in two (28.60%) of the deceased patients. Four patients presented with intestinal bleeding; one patient presented conjunctival bleeding; two patients presented with chest wall bleeding; one patient presented with bleeding at the puncture site. Bleeding was not the cause of death in any of these patients. Survival was not significantly affected by the presence of major bleedings (Figure 7). Of the seven deceased patients, one had a thrombotic event, and three had infection complications, which represented 75% of the infections that were observed. Survival did not differ between the patients with provoked or spontaneous bleeding (*p* = 0.30).

## 4. Discussion

This study presents data from a single-centre cohort, including the 26 patients treated in 15 years, which supports the rarity of the condition, with a variable incidence between 0.10 and 1.48 per million inhabitants per year [2,5,6,7,8,9,10,11,12,13,14,15,16]. Incidence was not calculated in our study because not all of the patients would have been referred to our tertiary centre and the referral population is not defined. Data present a similar overall AHA distribution in men and women; however, AHA was more frequent in male patients who were over 70 years old (77%), with a second peak being observed in the female patients aged 30–40 years old (19.3%), corresponding to postpartum AHA. AHA was not present in any male patient younger than 50 years old, which is consistent with the findings in large patient groups from the United Kingdom Haemophilia Centre Doctors’ Organisation (UKHCDO); the Acquired Hemophilia Working Group of the German, Austrian and Swiss Thrombosis and Hemostasis Society (GTH-AH); the French Surveillance des Auto antiCorps au cours de l’Hémophilie Acquise (SACHA) Registry; the US Hemostasis and Thrombosis Research Society; the European Acquired Haemophilia Registry (EACH2) [16,19,20,21,22].

Approximately half of our patients had no underlying condition, and malignancies were most often observed in the elderly patients, which is consistent with the distribution of underlying conditions published in previous studies; however, these studies observed that autoimmune diseases were the most common condition in young patients, a finding that differs from our study, which showed that pregnancy was the most common underlying condition in this age group (Table 5), a finding that is most likely due to the fact that the data were obtained from a single centre that is a reference obstetric centre with a high number of treated patients.

All of our patients presented at least one bleeding event, with haematomas being the most common (96% of the patients), most of which were mainly in the lower or upper limbs (41.30%). Thirty-three bleeding episodes were analysed, 33.33% of which had no other bleeding apart from the previously mentioned haematomas. Urinary tract bleeding was the second most frequent type of bleeding (27.70%), followed by intestinal (15.15%), intra-abdominal (9.09%), and vaginal (9.09%). Rare bleedings were observed in the central nervous system, postsurgical drainage area, oral mucosa, conjunctiva, and perineum (3.03% each). Diagnostic or therapeutic invasive procedures were a cause of provoked bleedings, although they were avoided in these patients during their management when it was possible, and the patients were supported by haemostatic treatment when required.

Bleeding patterns were similar in all of the patients with AHA, but the bleeding patterns differed from that of patients with congenital haemophilia A [5,7,28,29]. Spontaneous bleeding occurred in a high percentage of cases (80.8%).

More than half of the patients required red blood cell transfusions (61.5%). Half of the patients had major bleeding, regardless of sex, underlying disease, age group, inhibitor titre, or FVIII level at diagnosis, which is consistent with the findings from the UKHCDO and EACH2 [16,20].

All of the patients were treated with bypassing agents or Human FVIII as haemostatic therapy as part of the bleeding management. Recombinant porcine FVIII (rpFVIII) was not used since it was not available in Spain by the time of the treatment. To eradicate the inhibitor, our patients underwent immunosuppressive therapy as a first-line treatment with a steroid and rituximab-based regimen, as it has been shown to result in longer remissions and fewer infectious events [21,30,31,32,33]. The use of rituximab as a first-line therapy may draw attention since the guidelines propose it as a second-line treatment [18]. This can be explained because these guidelines were published in 2020 and our cohort comprises January 2006 and January 2021, when there was no consensus on the treatment of these patients, and treatment was chosen based on clinician experience.

Our patients responded well to the treatment; 84.60% of our patients eradicated the inhibitor, but four patients required the continuation of IST. A total of 69.20% of the cases met the criteria for CR. The eight patients who did not achieve complete remission were in the over-65 age group.

Definitions of remission vary in the large publications; the UK Surveillance study defined complete remission as normal factor VIII, undetectable inhibitor, and stopped immunosuppression or immunosuppression reduced to the doses used before the development of the acquired haemophilia without relapse [16]; in EACH2 CR is the clearance of the autoantibody and stable FVIII activity of >70 U dL^−1^ without replacement therapy; the term of “stable remission” is also included for cases which required the continuation of IST [17]. SACHA considered remission when the inhibitor titre was <0.6 BU mL^−1^, along with a normal FVIII:C level (>50 IU dL^−1^), and when no more bleeding symptoms were observed [23]. The GTH study also included a definition for partial remission (PR): FVIII restored to >50% and no bleeding after stopping any haemostatic treatment for at least 24 h, and used complete remission as PR plus an undetectable inhibitor (<0.6 BU/mL), prednisolone reduced to <15 mg/d (or equivalent glucocorticoid dose), and the discontinuation of any other IST [26]. Even so, a similar response has been reported with a rate of CR between 61 to 72% (Table 5), which are similar to the findings in our study. Spontaneous remissions have also been reported in other publications [10]; however, all of our patients received IST at presentation; therefore, a comparison is not possible.

Infections were similar to other series in 15% of the cases [16,17,23,26,27]; the increased infection risk observed in these patients might be due to treatment with immunosuppressants. Since treatment with immunosuppressants such as rituximab is needed for inhibitor eradication, we recommend evaluating the benefit-to-risk ratio to choose the more suitable strategy of immunosuppression in each patient. A total of 7.7% of the patients experienced thrombotic events, consistent with the findings of the SACHA and GTH [23,26]. Bypassing agents are considered contributing factors that increase the risk of these thrombotic events. In our study, seven patients died, five of whom were in the over-65 age group and had underlying malignancies; none of the deaths were associated with bleeding complications. IST could have increased the risk of infections, which were, along with malignancies, the main cause of death in our study (three patients in each case). The coronavirus disease 2019 (COVID-19) pandemic was ongoing at the end of the follow-up and affected one of the patients in our study.

## 5. Conclusions

AHA is a rare disease that increases the risk of major bleedings, affects men and women, and that has underlying conditions in half of the reported cases. Haemostatic and immunosuppressive treatments are essential for treating bleeding and eradicating the inhibitor. Hospital procedures are a cause of bleeding and should be closely monitored and avoided or delayed until after the inhibitor has been eradicated when possible. Cases of fatal infections are seen in AHA patients. The patients in this study received the treatments that were available to them. The emergence of new treatments for preventing bleeding, such as emicizumaband, the scientific evidence that has been published in recent years will change the approach to these patients to a more individualized management strategy. Early recognition, rapid diagnosis, and referral to an experienced centre with laboratory facilities are essential for the appropriate management of AHA with regard to bleeding risk and possible associated complications.

## Figures and Tables

**Figure 1 jcm-11-02721-f001:**
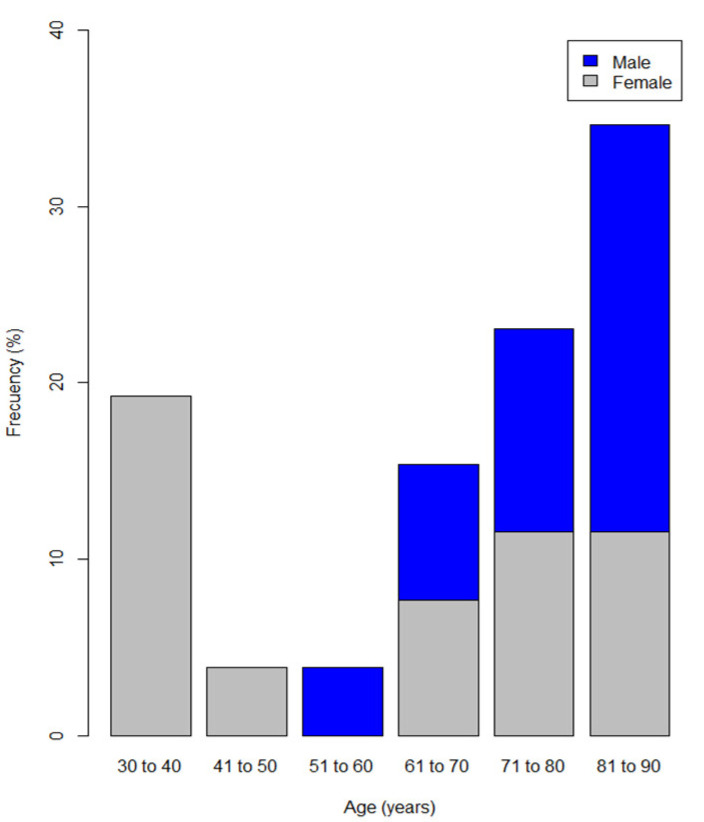
Distribution of patients diagnosed with acquired haemophilia A by age range and sex.

**Figure 2 jcm-11-02721-f002:**
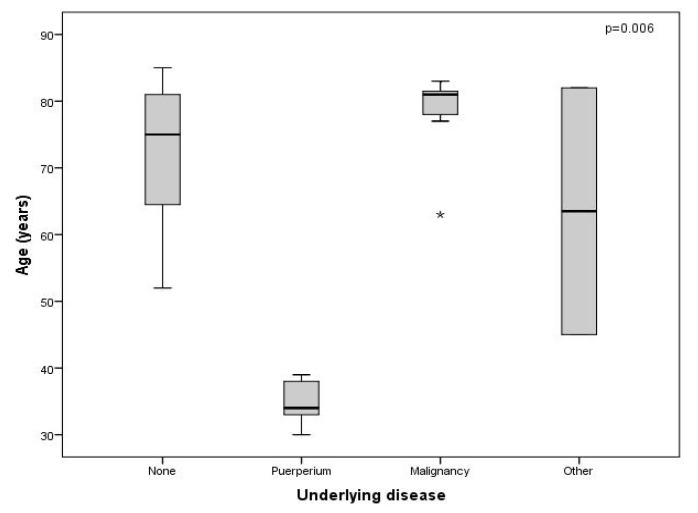
Distribution of patients diagnosed with acquired haemophilia A by underlying condition and age. Independent samples from non-parametric tests: Kruskal–Wallis. “None” is also described as idiopathic in the text. “Other” includes psoriatic arthritis and Paget’s disease. * Isolated case.

**Figure 3 jcm-11-02721-f003:**
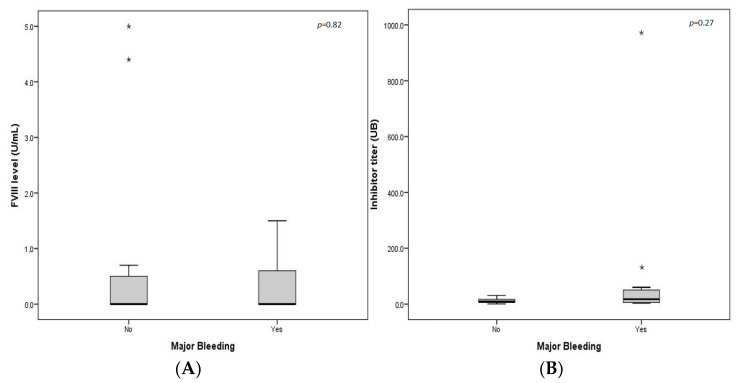
(**A**) Major bleeding episodes according to FVIII levels. (**B**) Major bleeding episodes according to inhibitor level. * Isolated case. Two independent samples from the non-parametric tests: U Mann–Whitney.

**Figure 4 jcm-11-02721-f004:**
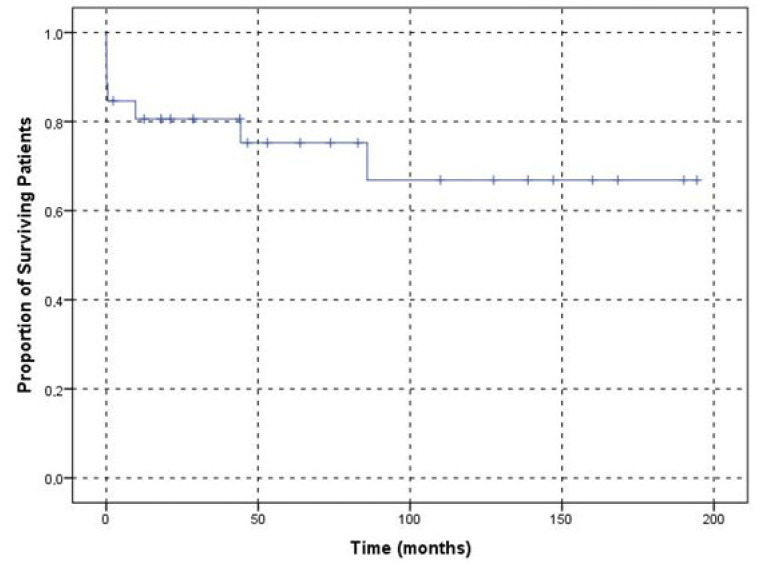
Kaplan–Meier plot showing the overall survival of patients with acquired haemophilia A.

**Figure 5 jcm-11-02721-f005:**
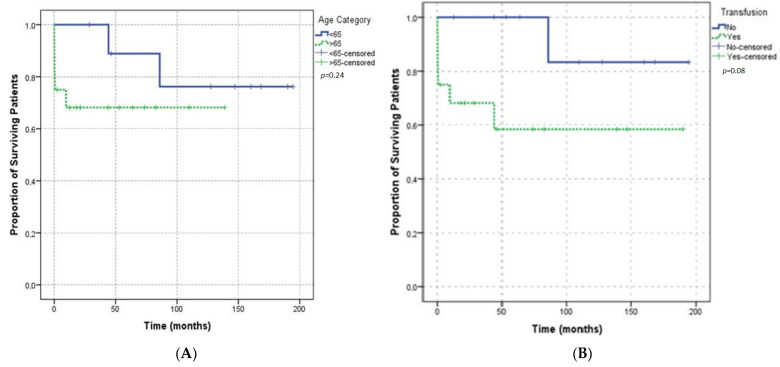
(**A**) Kaplan–Meier plot showing the overall survival according to age group. (**B**) Kaplan–Meier plot showing the overall survival according to transfusion requirements.

**Figure 6 jcm-11-02721-f006:**
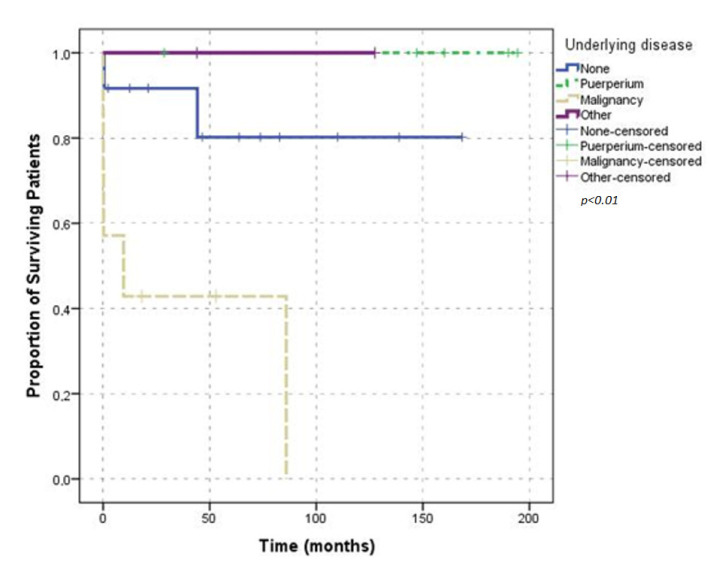
Overall survival according to underlying disease.

**Figure 7 jcm-11-02721-f007:**
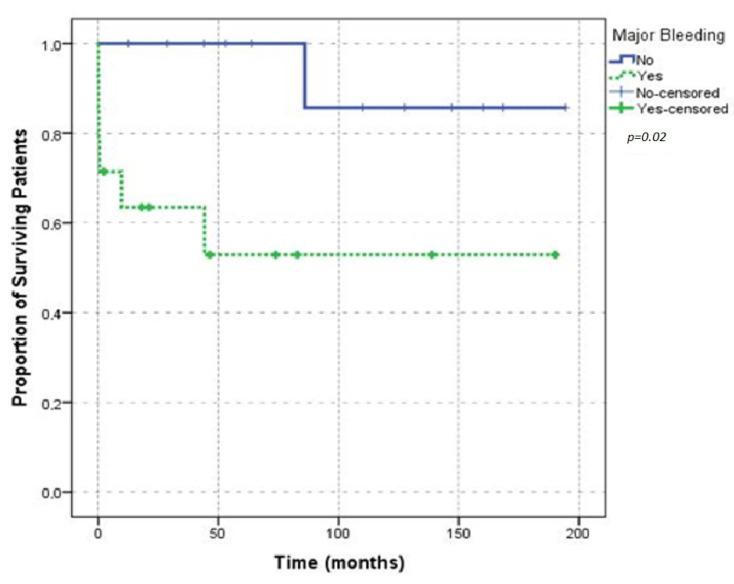
Overall survival according to major bleeding.

**Table 1 jcm-11-02721-t001:** Bleeding characteristics of the patients with acquired haemophilia A.

Characteristic	Frequency (%)
**Site of bleeding**	Total events 33 (100)
Haematomas only	11 (33.3)
Urinary	9 (27.3)
Intrabdominal	3 (9.1)
Perineal	1 (3.0)
Vaginal	3 (9.1)
Gastrointestinal	5 (15.2)
Subdural	1 (3.0)
**Haematoma distribution**	Total events 29 (100)
Generalised ecchymosis	1 (3.5)
Lower or upper limbs	12 (41.3)
Abdominal wall	3 (10.2)
Intrabdominal	4 (13.8)
Thoracic wall	3 (10.2)
Face	1 (3.5)
Perineum	1 (3.5)
Conjunctival	1 (3.5)
Oral mucosa	1 (3.5)
Central nervous system	1 (3.5)
Drainage zone	1 (3.5)
**Cause**	No. Of patients 26 (100)
Spontaneous	16 (61.6)
Provoked	5 (19.2)
Spontaneous and provoked	5 (19.2)
**Provoked events**	Total events 19 (100)
Puncture	6 (31.6)
Trauma	4 (21.1)
Gastric ulcer	2 (10.5)
Childbirth	3 (15.8)
Urinary catheter	1 (5.3)
Surgery	3 (15.8)

**Table 2 jcm-11-02721-t002:** Laboratory at diagnosis.

Laboratory		*n* (%)	Median	IQR
Haemoglobin (g/dL)		26 (100)	9.85	7.90–11.97
aPTT (s)		26 (100)	72.25	54.42–86.95
Inhibitor titre (BU/mL)	All	25 (96.15)	11.00	5.80–29.0
	0–10	12 (46.15)	5.25	4.20–7.70
	10–50	11 (42.30)	26.60	11.60–33.90
	>50	2 (7.69)	550	128–972
FVIII (U/dL)	All	26 (100)	0.85	0.5–1.5
	<1.0	21 (80.78)	0	0–0.1
	1.0–50	5 (19.23)	1.50	1.5–4.4

BU, Bethesda units; IQR, interquartile range; aPTT, activated partial thromboplastin time; FVIII, factor VIII.

**Table 3 jcm-11-02721-t003:** Major bleeding.

	Major Bleeding		*p*
	No	Yes	
**Underlying condition**			
Malignancy	2 (7.69)	5 (19.23)	0.098
Idiopathic	4 (15.38)	8 (30.76)	
Puerperium	4 (15.38)	1 (3.85)	
Other	2 (7.69)	0 (0%)	
**Age by group**			
<65 years	7 (26.92)	3 (11.53)	0.063
≥65 years	5 (19.23)	11 (42.30)	
**Sex**			
Female	9 (34.61)	5 (19.23)	0.267
Male	3 (11.54)	9 (34.62)	

Data are reported as *n* (IQR). Chi-squared test with Yates correction or Fisher’s exact test. “Other” includes psoriatic arthritis and Paget’s disease.

**Table 4 jcm-11-02721-t004:** Haemostatic treatments.

Treatment	*n* (%)	Doses (U)		
		Range	Median	IQR
**Recombinant FVIIa**	15 (57.70)	5–670	70.00	27–151
**aPCC**	3 (11.50)	42,000–150,000	124,000	*
**Human FVIII**	25 (96.10)	5000–722,000	111,000	63,000–205,000

IQR, interquartile range. * Not calculated for the number of events. FVIIa, activated recombinant Factor VII; aPCC, activated prothrombin complex concentrate; FVIII: Factor VIII.

**Table 5 jcm-11-02721-t005:** Demographic comparison with large patient groups with acquired haemophilia A.

	HULP	GTH-AH 01/2010 [26]	UKHCDO [16]	EACH2 [17]	SACHA [23]	HTRS [27]
** *n* **	26	102	172	501	82	166
	2006–2021	2010–2013	2001–2003	2003–2009	2001–2005	2000–2011
**Median age** **(years)**	66 (30–85)	74 (26–97)	78 (2–98)	74 (62–80)	76.70 (25–103)	65.30 (36–78)
**Sex (%)**						
**Female**	53.90	42	57	46	39	50
**Male**	46.10	58	43	53	61	50
**Underlying Condition (%)**						
**Idiopathic**	46.10	67	63.30	51.30	55	12.65
**Pregnancy**	19.30	5	3	8.8	6.10	3.40
**Malignancy**	27	13	14.60	12.30	19.50	14.50
**Autoimmune disease**	3.80	20	16.60	13.10	15	28.40
**Other**	3.80	ND	3.33	7.80	4.40	42
**Laboratory**						
**FVIII: C IU/dL**	1.50 (0.20–5)	1.40 (<1–31)	3 (1.70–7)	2 (* 1–5)	2 (0–30)	2
**Inhibitor** **titre BU/mL**	58.80 (0.70–972)	19 (1–1449)	13 (4–38)	12.80 (* 4.30–42.50)	16 (1–2800)	ND
**Hb (g/dL)**	9.85 (7.90–11.97	ND	ND	8.9 (* 7.30–11.10)	6.10 (3.10–12.70)	ND
**CR (%)**	69.20	61%	71%	72%	61	ND
**Mortality**	26.90	33%	43%	26%	33	ND

HULP, Hospital Universitario La Paz; GTH-AH 01/2010, Acquired Hemophilia Working Group of the German, Austrian and Swiss Thrombosis and Hemostasis Society 01/2010 study; UKHCDO, UK Haemophilia Centre Doctors’ Organisation; EACH2, European Acquired Hemophilia Registry; SACHA, Surveillance des Auto antiCorps au cours de l’Hémophilie Acquise Registry; HTRS, The Hemostasis and Thrombosis Research Society Registry; Data reported median (range); * IQR; ND, data not reported; Hb, haemoglobin; CR, complete remission. IU: international units; BU: Bethesda units; FVIII: Factor VIII.

## Data Availability

The data presented in this study are available on request from the corresponding author.

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
