# Peer review of "Acquired Haemophilia A: A 15-Year Single-Centre Experience of Demography, Clinical Features and Outcome"

_jcm, 2022, doi:10.3390/jcm11102721_

Round 1

Reviewer 1 Report

  • p3 line 108: which percentage of malignancy was found by screening (p 2; l 72: methods).
  • p7 ; line 194: abbrevation IST is not explained. Is it the same as ISP?

Author Response

Ammended as suggested. 

  • p3 line 108: which percentage of malignancy was found by screening (p 2; l 72: methods).

All the malignancies reported were presented by the moment of the AHA diagnosis, no new malignancy was found during the screening. 

  • p7 ; line 194: abbrevation IST is not explained. Is it the same as ISP?

Abreviation IST added, this replaces ISP. 

English editing service hired in the Journal website. 

Thank you.

Reviewer 2 Report

  • p3 line 108: which percentage of malignancy was found by screening (p 2; l 72: methods) and which percentage was known before diagnose hemophilia.
  • p7 ; line 194: abbrevation IST is not explained. Is it the same as ISP?

Author Response

Ammended as suggested. 

  • p3 line 108: which percentage of malignancy was found by screening (p 2; l 72: methods) and which percentage was known before diagnose hemophilia.

All the malignancies reported were presented by the moment of the AHA diagnosis, no new malignancy was found during the screening. 

  • p7 ; line 194: abbrevation IST is not explained. Is it the same as ISP?

Abreviation IST added, this replaces ISP. 

English editing service hired in the Journal website. 

Thank you.

This manuscript is a resubmission of an earlier submission. The following is a list of the peer review reports and author responses from that submission.

Round 1

Reviewer 1 Report

In their manuscript, the Authors described the characteristics of a small population of patients with acquired hemophilia A (AHA). Given the rarity of this condition, the information provided may be of interest.

However, 26 patients are too small a number to provide a simple descriptive analysis of the population. The data is poorly presented, with a short Introduction section, not clearly reporting the study objectives and already reporting part of the results (i.e. the number of participants). Similarly, the Discussion section is very poor, without any clear interpretation of the results.

First, given the simplicity and purely descriptive nature of the information provided, the manuscript should be resubmitted as a brief report or communication. Furthermore, the Introduction and Discussion sections should be extensively implemented. The Methods section should also be rewritten, detailing the inclusion criteria and any exclusion criteria, specifying that an Ethics Committee has expressed a favorable opinion for the retrospective data collection.

Most importantly, the Results section should be supported by more data and statistical analyses (e.g. correlations, regression analyses). In keeping with this, it would be important to have an initial table showing the main demographic and clinical characteristics of the study population, with any concomitant clinical condition at the time of AHA diagnosis.

Reviewer 2 Report

This study decribes the single centre patients characteristics with a acquired haemophilia A, presented between 2006 and 2021. They have described 26 patients. Furthermore comparrison with (international) other cohorts are made. Compared woth other studies pregnancy is a more common cause of acquired haemophilia A, the authors expect this to be due to the fact that the hospital is a obsteric reference center. 

As one of the conclusion is the need for swift referal to a experience center, why this experience is needed should be more evident. 

In that light I miss some essential information:

  • A distinction should be made if at presentation an underlying cause is known or is suspected. In my experience a new diagnosis of (occult) malignancy is rare in the setting of Acq HA. So extensive screening for an underlying malignancy, with risk for invasive procedures (eg incidentalomas).
  • More information on the type of factor concentrates (FEIBA / novoseven/ FVIII ) and IU used. As this will highlight the complexity of treatment of aqc HA. 
  • More details of the hospital procedures that caused a bleed. Was this something that could have been prevented? Eg: was the procedure before or after the diagnosis of acq HA. What kind of procedures? 
  • Information on the treatment for elimination of the inhibitor (steroids / rituximab etc).
  • Longer  term outcome. In other studies it is impressive that a high number of fatalities was due to a fatal infection. Underlining again the need for a expertise centrum.